# Improving Dietary Intake of Essential Nutrients Can Ameliorate Inflammation in Patients with Diabetic Foot Ulcers

**DOI:** 10.3390/nu14122393

**Published:** 2022-06-09

**Authors:** Raedeh Basiri, Maria Spicer, Cathy Levenson, Thomas Ledermann, Neda Akhavan, Bahram Arjmandi

**Affiliations:** 1Department of Nutrition and Food Studies, George Mason University, Fairfax, VA 22030, USA; 2Department of Nutrition and Integrative Physiology, Florida State University, Tallahassee, FL 32306, USA; mspicer@fsu.edu (M.S.); nsa08@my.fsu.edu (N.A.); barjmandi@fsu.edu (B.A.); 3Center for Advancing Exercise and Nutrition Research on Aging, Florida State University, Tallahassee, FL 32306, USA; 4Department of Biomedical Sciences, College of Medicine, Florida State University, Tallahassee, FL 32306, USA; cathy.levenson@med.fsu.edu; 5Department of Family and Child Sciences, Florida State University, Tallahassee, FL 32306, USA; tledermann@fsu.edu

**Keywords:** nutrient supplementation, nutrition education, diabetes, diabetic foot ulcer, chronic wounds, pro-inflammatory cytokines, anti-inflammatory cytokines, nutrition intervention, inflammation, wound healing, cytokines, CRP, IL6, IL10, tristetraprolin, TTP

## Abstract

Diabetic foot ulcers (DFUs) are classified as chronic wounds and are one of the most common complications of diabetes. In chronic wounds, management of inflammation is a key step in treatment. Nutrition plays an important role in managing and controlling inflammation. This study evaluated the effects of nutrition supplementation and education on inflammatory biomarkers in patients with DFUs. Eligible patients with foot ulcers were randomly assigned to either a treatment (*n* = 15) or control group (*n* = 14). Both groups received standard care for wound treatment from the clinic; however, the treatment group was also provided with nutritional supplementation and education. Plasma concentrations of inflammatory biomarkers, namely C-reactive protein (CRP), interleukin 6 (IL6), interleukin 10 (IL10), and tristetraprolin (TTP), were evaluated at baseline and every four weeks, until complete wound closure had occurred or up to 12 weeks. The mean plasma concentration of IL6 significantly decreased in the treatment group (*p* = 0.001). The interaction between time and group was not statistically significant for the mean plasma concentrations of CRP, IL10, and TTP during the 12 weeks of the study. The results of this study showed the positive effects of nutritional intervention on controlling inflammation in DFU patients. More clinical trials with a larger population and longer duration of time are needed to confirm our results.

## 1. Introduction

Diabetic foot ulcers (DFU) are one of the most common causes of lower extremity amputation in diabetic patients [1]. About a quarter of people with diabetes will develop a foot ulcer, and up to 16% of DFUs will lead to amputation if they go untreated [2]. Wound healing is a complex process, including a series of interactions between various cell types, extracellular matrices, and cytokine mediators. Inflammation is a typical physiologic response to tissue injury, which is essential for disinfecting the wound area and tissue repair processes [3]. In normal wounds, acute inflammatory responses continue for a short duration of time and resolve promptly due to negative feedback mechanisms; however, in chronic wounds such as DFUs, inflammatory responses fail to regulate themselves, which results in chronic inflammation and deterioration of the healing process [4]. In DFU patients, hyperactivity of inflammatory cells results in a higher generation of reactive oxygen species (ROS), which, along with downregulation of anti-inflammatory factors such as interleukin 10 (IL10), will add to the burden of inflammation [5]. A high concentration of ROS can degrade the extracellular matrix by increasing the expression of matrix metalloproteinase (MMPs) and increasing the production of other free radicals, which worsens the healing process [6,7]. In addition to the high production of ROS in diabetes, there is a diminished ability to remove them due to defective reducing complexes (glutathione), reducing enzymes (glutathione reductase), and reducing amino acids (cysteine) [5]. A combination of hyperglycemia as well as hypoxia, caused by disrupted vasculature in diabetic patients, limits wound healing and inhibits neutrophil and macrophage function which increases the risk of infection [8,9]. Additionally, upregulation of proinflammatory cytokines interleukin 1β (IL1β), interleukin 6 (IL6), and tumor necrosis factor-alpha (TNF-alpha), alongside down-regulation of anti-inflammatory molecules, such as transforming growth factor-beta (TGF-β) and IL10, result in a chronic non-healing wound in diabetic patients [10,11]. Therefore, controlling inflammation and infection within and surrounding the wound site is a major goal in DFU wound care. IL6 and C-reactive protein (CRP) are among the best indicators of inflammation and wound healing in DFU patients. IL6 expression is strongly correlated with both glucose concentration and wound chronicity [12]. A high concentration of IL6 is also a promising predictor of delayed wound healing and infection in DFU patients [13,14]. Additionally, IL6 and CRP are both correlated with the size of the wound in patients with DFUs [15]. A study by Weigelt et al. showed that patients with grade 3 DFUs, according to the University of Texas Wound Classification [16], had significantly higher blood levels of CRP and IL6 compared to those with grade 1 [15]. Decreased expression of anti-inflammatory biomarkers, such as IL10, was also observed in keratinocytes and endothelial cells at the wound margins in DFU patients (*p* < 0.05) [17]. Evidence has shown that a low concentration of IL10 contributes to the development of non-healing wounds in diabetic patients [18]. Inhibition of inflammatory biomarkers increases the expression of anti-inflammatory biomarkers, such as IL10, which is essential for promoting wound healing [19,20]. Various dietary components, including antioxidant vitamins and minerals, have the potential to alleviate chronic inflammation and could play a key role in chronic-wound healing. Vitamins C, A, and E, as well as zinc, manganese, and copper, are strong antioxidants and show potent anti-inflammatory effects [21,22,23,24,25,26,27,28]. Therefore, it is essential to examine if these nutrients can shift diabetic wounds from chronic/non-healing to normal wounds by improving inflammation status. It has been reported that DFU patients have a significantly low intake of the aforementioned nutrients [29,30]. However, evidence of the benefit to wound healing in DFUs through management of inflammation is lacking for a combination of these essential nutrients. Thus, further studies addressing the efficacy of these nutrients in controlling inflammatory conditions linked to DFUs are required. The effectiveness of utilizing educational tools for improving blood glucose and foot self-care behavior in patients with type 2 diabetes has been reported [31,32]; however, to our knowledge the effects of nutrition education in improving inflammatory biomarkers in patients with DFUs have not been evaluated yet. This study provides valuable information about the effects of supplementing with a glucose control formula, consisting of essential nutrients for wound healing, as well as nutrition education on inflammatory biomarkers in patients with DFUs. This study hypothesized that improving dietary intake of antioxidants through nutritional supplementation and nutrition education would improve the inflammation status related to DFUs. 

## 2. Materials and Methods

### 2.1. Sample Size

The sample size was calculated based on an effect size of Cohen’s *d* = 0.25 (standardized mean treatment difference) for the primary outcome measure (reduction in IL6 at 12 weeks), which would be a clinically meaningful change [33]. Using G*Power statistical software, and in order to have 80% power for detecting this difference with an α error of 0.05, we needed at least 24 patients. Anticipating a 10–20% dropout rate, the target sample size was 27–29, with 14–15 patients in each group (control, treatment).

### 2.2. Screening and Recruitment

This study was approved by the Institutional Review Board (IRB) of both Florida State University and Tallahassee Memorial Hospital (TMH, Tallahassee, FL, USA). The trial is also registered at clinicaltrials.gov(Identifier:NCT04055064). Due to the requirement of IRB regarding confidentiality of personal health information, a point of contact from the clinic (a nurse or one of the medical staff) was identified. They were educated about the study and informed of the inclusion/exclusion criteria. They explained the study and provided flyers to potential participants. Individuals who were willing to participate in the study were then referred to the researcher for further screening. Inclusion criteria were non-pregnant/non-lactating females or males between the ages of 30 and 70 years with type 1 or 2 diabetes. Participants must have been undergoing pharmacological treatment for glycemic control, with at least one foot ulcer of grade 1A based on the University of Texas classification [16]. 

Patients were excluded from the study if they had: used bioengineered tissue within four weeks prior to baseline; a history of radiation treatment to the ulcer site; hemoglobin A1c (HbA1c) > 12%; known immunosuppression; active malignancy; chronic kidney disease; liver failure/cirrhosis; or heart failure and/or myocardial infarction in the past three months. Additionally, excessive use of alcohol based on the World Health Organization standards, current use of warfarin, or any mental or physiological condition that may interfere with supplement intake warranted exclusion from the study. 

After screening, eligible patients were informed about the details of the study and consent was obtained from patients who were interested in participating. This study was a randomized control trial with repeated measures. Participants were randomly assigned to either the treatment or the control group. All participants were consecutively enrolled in the study and followed up until the time that complete wound closure had occurred or up to 12 weeks, whichever came first. 

### 2.3. Intervention/Treatment

All participants, irrespective of their grouping, received standard wound care from the TMH wound care clinic. Additionally, participants in the treatment group were instructed to consume two servings (474 mL) of Boost Glucose Control (Nestle Health Science, Bridgewater Township, NJ, USA), a proprietary produced glucose control nutritional formula (supplement) between meals, preferably one in the morning and one in the afternoon, for 12 weeks or until complete wound closure occurred. Participants in the intervention group were also educated about improving their dietary intake by increasing their consumption of low-fat/high-bioavailable protein sources, vegetables, and high-fiber carbohydrates, as well as decreasing their intake of refined and simple carbohydrates. Nutrition education was conducted in-person by the primary researcher (nutritionist) for 10 min at baseline. Participants were then reminded about the educational materials and were given a chance to ask questions during following visits until complete wound closure had occurred or up to 12 weeks. Consuming two servings of the supplement provided patients in the treatment group with a total energy of 500 calories, 28 g of protein, and essential vitamins and minerals for wound healing. A complete list of the nutrient content of the supplement is shown in Appendix A. 

We aimed to provide patients with a supplement that could deliver at least 50% of the RDA recommendation for essential nutrients. We anticipated that nutrition education would improve the dietary intake of nutrients and motivate patients to meet the remaining 50% of nutrient recommendations by consuming better food sources. Table 1 shows the recommended daily allowance (RDA) for nutrients involved in wound healing and compares it with the nutrient content of two servings of the supplement.

### 2.4. Data Collection

At baseline, all participants were asked about medical and medication history. Anthropometric measurements, dietary assessments, and blood sample collections were conducted at baseline and repeated every four weeks until complete wound closure had occurred or up to 12 weeks.

### 2.5. Blood Sampling and Processing

A finger stick blood collection was conducted to assess HbA1c using HbA1c Now + test (Polymer Technology Systems, Indianapolis, IN, USA). Venous blood samples were collected from an antecubital vein using a vacutainer brand collection set. Blood samples were collected for conducting enzyme-linked immunosorbent assay (ELISA) tests on inflammatory biomarkers. Blood samples were obtained at initial assessment/baseline, at 4, 8, and 12 weeks, or at the time of complete wound closure. These were taken at the wound care clinic and were then transported to the Florida State University (FSU) research lab, following the Centers for Disease Control (CDC) guidelines for transporting blood samples. In the FSU research lab, specimens were centrifuged, separated, and aliquoted into labeled tubes, and stored in a −80 °C freezer until needed for analysis.

### 2.6. Plasma Preparation

After collecting whole blood into anticoagulant tubes, cells were removed from the plasma by centrifugation using an IEC CL31R multispeed refrigerated centrifuge (Thermo Electron Corporation, Waltham, MA, USA) at 1500× *g* for 10 min [34]. Following centrifugation, plasma was immediately transferred into 0.5 mL aliquots and was stored at −80 °C until analysis. 

### 2.7. Biochemical Analysis

The inflammatory biomarkers IL6, IL10, CRP, and TTP were examined to evaluate the effects of the intervention on inflammation in DFUs. Evaluation of IL6 and CRP was undertaken using a human C-Reactive Protein/CRP Quantikine ELISA Kit and a human IL6 Quantikine ELISA Kit from R&D systems (Biotechne, Minneapolis, MN, USA). For CRP, the mean CV% for intra-assay precision was 5.5% and for inter-assay precision was 6.5%. The reported mean CV% for IL6 for intra-assay precision was 2.6% and for inter-assay precision was 4.5%. Assessments of IL10 and TTP were conducted using human ELISA kits from MyBioSource (San Diego, CA, USA). The mean CV% for intra-assay and inter-assay for IL10 were 4.44% and 6.27%, respectively, and for TTP were ≤8% and ≤12%, respectively. We were not able to read the data following standard instructions for IL10 and TTP ELISA tests, since our population had an exceptionally low concentration of these biomarkers. After consulting with the company’s technical support, when evaluating IL10 using ELISA kits the first incubation time was increased to 2 h and the plate was put in a slow shaker during incubation. We also performed the test with standard curve assayed in serum diluent, and samples assayed and incubated at 4 °C overnight (20 h). No other changes were made to other procedures and protocols. 

### 2.8. Statistical Analysis

Data were analyzed using the Statistical Package for Social Science (SPSS) version 25.0 (IBM SPSS). The statistical significance value was set at *α* < 0.05 for all tests. Descriptive statistics and independent-sample *t*-tests were used to compare means of potential confounding variables between groups at baseline. Multilevel modeling (mixed model) was used for the analysis of inflammatory biomarkers. The effects of potential confounding factors for different variables were examined and if the effect was significant, they were added as covariates to the model. If a significant F-statistic was obtained, Bonferroni’s post hoc test was used for pairwise comparisons.

## 3. Results

### 3.1. Baseline Characteristics

In total, 95 patients were screened, but only 42 met the inclusion criteria and were willing to participate in the study. Overall, 13 participants were excluded from the study due to a change in their clinic. Therefore, clinical, laboratory, and statistical analyses was performed on a total of 29 patients. 

### 3.2. General Characteristics

Both groups had similar characteristics at baseline. The general characteristics of participants at baseline has been shown in Table 2.

The average age of the study population was 53.3 ± 11.1 years (mean ± SD). There were no statistically significant differences between participants regarding the duration of diabetes, estimated wound age, HbA1c, ethnicity, age, or body mass index (BMI) at baseline. Although the distribution of gender was different in the treatment and control groups, the effect of gender on each variable was examined and added to the model as a covariate if the effect was significant. The mean duration of diabetes was greater in the treatment group (14.4 ± 8 year) in comparison with the control group (11.7 ± 6 year); however, the difference was not statistically significant. There were no significant differences between the groups regarding indicators of socioeconomic status (SES) or other factors that could potentially affect the dietary intake of participants, including appetite problems or religious and cultural restrictions. Living, financial, and employment status, as well as a need for food assistance, were considered indicators of SES. 

### 3.3. Changes in Plasma Concentrations of Inflammatory and Anti-Inflammatory Biomarkers during the Study Period

#### 3.3.1. Changes in Plasma Concentrations of IL6 and CRP

There were no significant differences between the plasma concentrations of IL6 in the treatment and control groups at baseline (16.1 pg/mL vs. 15.8 pg/mL, respectively). The potential effects of confounding factors such as duration of diabetes, estimated wound age (*p* = 0.004), gender, HbA1c at baseline, age, smoking, and BMI were examined. Only estimated wound age had a statistically significant effect on the concentrations of IL6, therefore this was used as a covariate in the model. The mean concentration of IL6 significantly decreased in the plasma of the treatment group after adjustment of estimated wound age (*p* = 0.001). In the control group, the mean concentration of IL6 was 15 times higher than its concentration in the treatment group at the end of the study. Comparison of the mean concentrations of IL6 at different time-points of the study for the treatment and control groups are outlined in Figure 1. 

There was no statistically significant difference between the mean concentration of CRP among the two groups at baseline. We examined the effects of potential confounding factors, such as HbA1c at baseline, age, gender (*p* = 0.02), smoking, estimated wound age (*p* < 0.001), duration of diabetes (*p* = 0.04), and BMI. The significant confounding factors (gender, estimated wound age, and duration of diabetes) were kept in the model for further analyses. The interaction between group and time was not statistically significant after adjustment of confounding factors.

#### 3.3.2. Changes in Plasma Concentrations of IL10 and TTP

There was no statistically significant difference between the mean concentration of IL10 among the two groups at baseline. The effects of potential confounding factors, such as HbA1c at baseline, age, gender, smoking, estimated wound age, duration of diabetes, and BMI, were evaluated; however, none of these factors had a significant effect on the plasma concentration of IL10, and, therefore, they were removed from the model. The interaction between group and time was not statistically significant for IL10.

Similar to IL10, the effects of potential confounding factors on the concentrations of the TTP were evaluated. Only HbA1c at baseline (*p* = 0.03) and BMI (*p* = 0.01) yielded significant on the concentrations of TTP. Additionally, the effect of gender on the concentration of TTP tended to be significant (*p* = 0.06); therefore, we included these factors as covariates in the model. The mean plasma concentration of TTP in the treatment group was increased numerically from 361 pg/mL at baseline to 1243 pg/mL at the end of the study. In the control group, the mean plasma concentration of TTP increased from 355 pg/mL to 479 pg/mL. Although the interaction between group and time was not statistically significant after adjustment of the confounding factors, the mean increase in the plasma concentrations of TTP in the treatment group was about seven times higher than the control group. Please see Figure 2.

## 4. Discussion

Our results showed that nutritional supplementation, used concurrently with nutrition education, can strongly decrease the plasma concentrations of IL6 in DFU patients. In contrast, the plasma concentrations of IL6 were increased drastically in our control group at the end of the study. Evidence has shown that an increase in IL6 could be an indicator of wound infection [13]; however, we did not collect data on wound infections to confirm this in our population. 

The plasma concentration of TTP increased numerically in our intervention group almost seven times more than the control group. TTP is an RNA binding protein that enforces the degradation of mRNA encoding cytokines and chemokines [35], and, therefore, reduces systemic inflammation through the under-expression of inflammatory mediators, including IL6, TNF alpha, and IL18 [36,37,38]. Additionally, TTP negatively regulates NF-κB signaling at the transcriptional corepressor level, which represses inflammatory gene transcription [39]. Although the interaction between time and group was not statistically significant for plasma concentrations of TTP, the observed numerical increase in TTP, along with a strong significant decrease in the concentrations of IL6 in our treatment group, might suggest positive effects of our intervention on controlling inflammation status via regulation of TTP in patients with DFUs. The positive effects of the intervention on increasing (numerically) the plasma concentration of TTP might also suggest the potential of a positive effect of our intervention on improvement of plasma concentrations of TTP during long-term application. These results are supported by other studies where they showed that serum and urinary levels of IL6 and IL18 were significantly elevated while TTP was significantly depressed in patients with diabetes [36]. We are not aware of a similar study which evaluates the effects of dietary supplementation or quality of diet on the concentrations of TTP. More clinical trials with a larger population and longer duration of time are needed to validate our findings. 

Our findings are consistent with the results of a study by Afzali et al., which showed that nutritional supplementation for 12 weeks reduced inflammation in patients with DFUs [40]. Similarly, another study showed that nutritional supplementation could reduce rate of infection and antibiotic use in patients with DFUs [41]. We are not aware of any other study that examines the effects of dietary supplementation on inflammatory biomarkers in DFU patients. However, it has been reported that, in general, high-quality diets have beneficial effects on reducing concentrations of inflammatory biomarkers in various chronic conditions. An inverse association between intake of whole grains, fruits, nuts, and green leafy vegetables, and concentrations of IL6 and CRP, has been reported by several studies [42,43,44,45]. Therefore, providing nutrition education as part of the intervention in this study could have had synergistic effects on observed improvement of inflammation status in the intervention group. The strength of our study was to use both nutrition education and supplementation for improving inflammation status in patients with DFUs. To our knowledge, this is the first study to evaluate the effects of dietary intake of nutrients on concentrations of TTP as an inhibitory factor for inflammation. This anti-inflammatory protein could serve as the basis for novel approaches to chronic wound therapy. 

Due to the relatively small sample size, we were not able to examine the effects of supplementation or education on inflammation status separately. Additionally, the potential effects of different blood-glucose-lowering medications, as well as the physical activity levels of the participants, were not evaluated. It has been reported that accumulation of advanced glycation end products (AGEs) contribute to DFUs [46,47]; however, we did not collect data on AGEs. Evidence has shown that inositol could decrease HbA1C in overweight patients with type 1 diabetes [48]. Moreover, it has been shown that inositol can reduce inflammation and oxidative stress related to diabetes in pregnant women with gestational diabetes [49]. Our supplement contained 200 mg of inositol per serving, which might have played a role in the observed improvement in inflammation status. Currently, there is no RDA for inositol. More studies need to be conducted to discover the optimum dosage of inositol for improving inflammation status in patients with DFUs. Analysis of dietary intake of our participants at baseline showed that our population had a significantly low intake of potent antioxidants such as vitamin E, vitamin A, zinc, copper, and manganese when compared with RDA recommendations; these data have been published elsewhere [50]. Presently, there are no recommendations for dietary intake of micronutrients/antioxidants for people with DFUs. Our aim in this study was to support participants in receiving an adequate amount of antioxidants based on the RDA recommendation; however, patients with DFUs might benefit from intakes above the RDA recommendation due to high production of ROS and the presence of an open wound. More clinical studies are needed to discover the optimum amount of each of these nutrients for improving inflammation status and wound healing in DFUs. In order to support the added benefits of other nutrients found in various foods, priority should be given to nutrition education to increase the quality of diet in this population. However, several studies showed that DFU patients could not receive an adequate amount of the essential nutrients from their food for several reasons [30,50,51,52]. When receiving enough nutrients from the diet is not possible, supplementing with optimum dosage of essential nutrients for improving conditions is recommended. Results of our study showed that educating patients with DFUs to consume high-quality diets, as well as providing them with supplements that support at least 50% of the RDA recommendation for antioxidants, have significant positive effects on improving inflammatory conditions. Currently, nutritional interventions or referral to dietitians are not part of standard care; however, as our results show, nutritional interventions are critical components of the wound healing process and should be an integral part of the treatment of DFUs.

## 5. Conclusions

Our findings showed that nutritional supplementation, along with nutrition education could significantly improve inflammation status in patients with DFUs. More clinical trials with larger sizes are needed to confirm our results.

## Figures and Tables

**Figure 1 nutrients-14-02393-f001:**
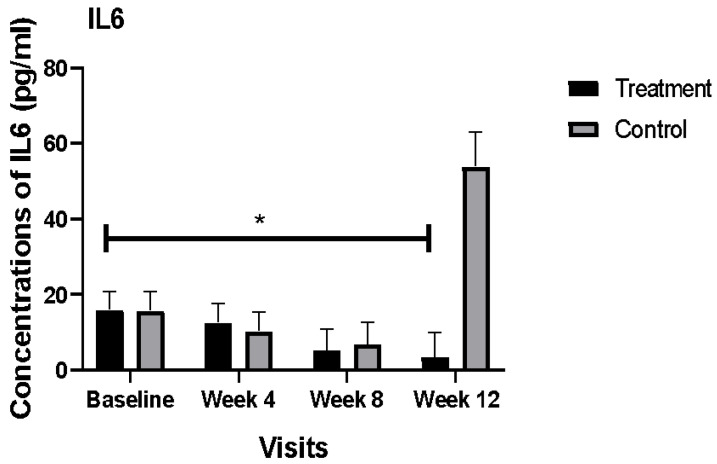
Comparison of mean plasma concentrations of IL6 (pg/mL) between the treatment and control group during the 12 weeks of the study. Bars represent the means ± SEM. * Denotes a significant time by group interaction (*p* < 0.05).

**Figure 2 nutrients-14-02393-f002:**
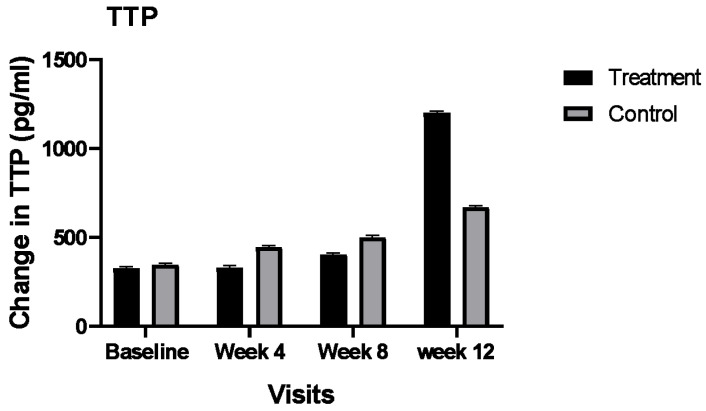
Comparison of mean plasma concentrations of TTP (pg/mL) between the treatment and control group during the 12 weeks of the study. Bars represent the means ± SEM.

**Table 1 nutrients-14-02393-t001:** Comparison of content of two servings of supplement vs. RDA recommendations for antioxidants for age and gender.

Nutrient	RDA for Nutrient	Total from Two Supplements/Day	% of RDA Provided; Men vs. Women if RDA Varied
Vitamin C	60	204	304%
Vitamin A	3000 IU	2500 IU	83%
Vitamin E	33.3 IU	66 IU	200%
* Manganese Men/Women	2.3/1.8 mg	0.8 mg	35%/44%
Copper	0.9 mg	0.8 mg	88%
Zinc Men/Women	11/8 mg	6 mg	54%/75%
Protein Men/Women	56 g/46 g	28 g	50%/61%

* No established RDA for Manganese, numbers are showing adequate intake (AI). IU: international unit.

**Table 2 nutrients-14-02393-t002:** Baseline characteristics of participants by group.

Groups	Treatment (*n* = 15)	Control (*n* = 14)	*p*-Value
Men/women	8/7	11/3	0.08
Age (year) Means ± SD	52.9 ± 9.74	53.8 ± 12.8	0.84
Ethnicity African American/white	4/11	3/11	0.75
BMI ^1^ (kg/m^2^) Means ± SD	33.5 ± 7.98	34.1 ± 6.04	0.84
Diabetes Duration (year) Means ± SD	14.40 ± 8.03	11.7 ± 6.17	0.32
Estimated wound age (m) Means ± SD	10.97 ± 15.09	10.58 ± 18.27	0.95
HbA1c ^2^ Means ± SD	7.95 ± 2.06	8.40 ± 2.16	0.57
Smoking (yes/no)	3/12	3/11	1.00

^1^ BMI: Body Mass Index ^2^ HbA1C: Hemoglobin A1c.

## Data Availability

The datasets generated from this study are available from the corresponding author upon reasonable request.

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
