# Peer review of "Improving Dietary Intake of Essential Nutrients Can Ameliorate Inflammation in Patients with Diabetic Foot Ulcers"

_nutrients, 2022, doi:10.3390/nu14122393_

Round 1

Reviewer 1 Report

This is an interesting work about the role of dietary intake of essential nutrients on  inflammation in subjects with diabetic foot ulcers. I have some suggestions and comments.

Introduction

  • As reported in abstract, about education treatment, could be useful to treat this role in introduction. See and use: DOI: 10.1007/s42000-018-0005-9; doi: 10.1016/j.diabres.2019.05.003.

Materials and methods

  • have you calculated the sample size?
  • The subjects were consecutively enrolled?
  • could you clarify how was performed educational treatment?

Results

  • Have you analyzed the role of concomitant anti hyperglycemic drugs?
  • data about any type of physical activity?
  • Have you data about Advanced glycation end-products?

Discussion

  • Have you data about inositol? You can see and use doi: 10.1007/s00592-016-0954-x
  • Please, see and use this about the role of AGE products: DOI: 10.1097/01.ASW.0000792932.01773.d5
  • Please, clarify better strength and limitations of your study. 

Author Response

Dear Editor:

We would like to thank the reviewers for taking the time and effort to review our manuscript. Our responses to the Reviewers’ comments are described below point by point. Changes, suggested by the reviewers, have been incorporated into the manuscript using track changes within the document. The responses are also provided below.

Reviewer 1:

  • As reported in abstract, about education treatment, could be useful to treat this role in introduction. See and use: DOI: 10.1007/s42000-018-0005-9; doi: 10.1016/j.diabres.2019.05.003. Thank you for bringing this up to our attention. We added the nutrition education treatment in the Introduction, lines 82 to 86, and cited the suggested publications. Please see the paragraph below.

“The effectiveness of utilizing educational tools for improving blood glucose and foot self-care behavior in patients with type 2 diabetes has been reported [31,32]; however, to our knowledge the effects of nutrition education in improving inflammatory biomarkers in patients with DFU have not been evaluated yet.”

Materials and methods

  • Have you calculated the sample size? We thank the reviewer for this comment. The sample size and power calculations have been added to the manuscript under the Materials and Methods section, lines 92 to 98. Please see the paragraph below.

Sample Size

The sample size was calculated based on an effect size of Cohen’s d = 0.25 (standardized mean treatment difference) for the primary outcome measure (reduction in IL6 at 12 weeks), which would be a clinically meaningful change [33]. Using G*Power statistical software, in order to have an 80% power for detecting this difference with an α error of 0.05, we needed at least 24 patients. Anticipating a 10-20% dropout rate, the target sample size was 27-29 with 14-15 patients in each group (control, treatment).”

  • The subjects were consecutively enrolled? Yes, the subjects were consecutively enrolled. We have added this in the Materials and Methods, lines 121 to 122. Please see the paragraph below.

“All participants were consecutively enrolled in the study and followed until the time that complete wound closure occurred or up to 12 weeks, whichever came first.”

  • Could you clarify how was performed educational treatment? The nutrition education was done in person by the primary researcher. We have added it to the nutrition education part under the Intervention/Treatment, line 134, to clarify this. Please see the paragraph below.

“Nutrition education was done in-person by the primary researcher (nutritionist) for 10 minutes at baseline. Participants then were reminded about the educational materials and were given chance to ask questions during the following visits until the complete wound closure occurred or up to 12 weeks.”

Results

  • Have you analyzed the role of concomitant anti hyperglycemic drugs? We agree with the reviewer that the use of different medications could affect the results. Patients who were using medication for controlling blood glucose were eligible for our study; however, due to the small sample size we did not analyze data for the effects of different medications. This has been added to the limitations of the study under the Discussion, lines 310 to 312. Please see below.

“Also, the potential effects of different blood glucose lowering medications as well as the physical activity levels of the participants were not evaluated.”

  • Data about any type of physical activity? Unfortunately, we did not collect data on the physical activity levels of participants. We have added this to the limitations of the study under the Discussion, lines 310 to 312.
  • Do you have data about advanced glycation end-products? We agree that increased advanced glycation end products contribute to DFU, but we did not collect data on the AGE products. We have added this to the limitations of the study under the Discussion, lines 312 to 314.

“It has been reported that accumulation of advanced glycation end products (AGEs) contributes to DFU [46,47]; however, we did not collect data on AGEs.”

Discussion

  • Do you have data about inositol? You can see and use doi: 10.1007/s00592-016-0954-x. We have added more information about inositol in the Discussion, lines 315 to 320, and cited the suggested references. Please see the paragraph below.

“Evidence has shown that inositol could decrease HbA1C in overweight patients with type 1 diabetes [48]. Moreover, it has been shown that inositol can reduce inflammation and oxidative stress related to diabetes in pregnant women with gestational diabetes [49]. Our supplement contained 200 mg of inositol per serving which might have played a role in the observed improvement in inflammation status. Currently, there is no RDA for inositol. More studies need to be done to discover the optimum dosage of inositol for improving inflammation status in patients with DFU.”

  • Please, see and use this about the role of AGE products: DOI: 10.1097/01.ASW.0000792932.01773.d5. This has been added to our limitations in the Discussion, lines 312 to 314. Please see below.

“It has been reported that accumulation of advanced glycation end products (AGEs) contributes to DFU [46,47]; however, we did not collect data on AGEs.”

  • Please, clarify better strength and limitations of your study. Surely, we added strength of our study as well as suggested limitations to the Discussion, lines 304 to 320. Please see the paragraph below.

The strength of our study was to use both nutrition education and supplementation for improving inflammation status in patients with DFU. To our knowledge, this is the first study to evaluate the effects of dietary intake of nutrients on concentrations of TTP as an inhibitory factor for inflammation. This anti-inflammatory protein could serve as the basis for novel approaches to chronic wound therapy. Due to the relatively small sample size, we were not able to examine the effects of supplementation or education on inflammation status separately. Also, the potential effects of different blood glucose lowering medications as well as the physical activity levels of the participants were not evaluated. It has been reported that the accumulation of advanced glycation end products (AGEs) contributes to DFU [46,47]; however, we did not collect data on AGEs. Evidence has shown that inositol could decrease HbA1C in overweight patients with type 1 diabetes [48]. Moreover, it has been shown that inositol can reduce inflammation and oxidative stress related to diabetes in pregnant women with gestational diabetes [49]. Our supplement contained 200 mg of inositol per serving which might have played a role in the observed improvement in inflammation status. Currently, there is no RDA for inositol. More studies need to be done to discover the optimum dosage of inositol for improving inflammation status in patients with DFU.”  

Reviewer 2 Report

Diabetic foot ulcer is a chronic wound characterized by inflammation. Inflammatory biomarkers are increased in diabetic foot ulcer, as indicators of wound infection.

The present study was well planned and performed and the results well presented. Although the number of participants in the study is small, the results showed that the mean plasma concentration of the biomarker interleukin 6 significantly decreased in the treatment group, in comparison to the control group. These results are indicative that nutrition supplements, in combination with nutrition education, have positive effect on improving inflammation in patients with diabetic foot ulcer. The supplementation with nutrients is very important for diabetic patients, leading to the reduction of infection, to a better wound healing and to the reduction of antibiotic use. Although the results of this study indicate the beneficial action of nutritional supplements and especially antioxidants to diabetic foot ulcer, more studies are needed to prove this beneficial action in the amelioration of diabetic foot ulcer, concerning more inflammatory biomarkers and not only interleukin 6.

Author Response

Dear reviewer,

Thank you for your feedback and positive evaluation of our study!